# Untangling iron threads: A deep dive into plant intracellular pools

Alexandra Leskova, Tou C. Xiong, Stéphane Mari and Catherine Curie ⓘD

IPSiM, University of Montpellier, CNRS, INRAE, Institut Agro, Montpellier, France

inter-organelle flux; ion quantification; iron pool; metal sensor; single cell analysis.

**Corresponding author**:
Catherine Curie;
Email: catherine.curie@cnrs.fr

A.L. and T.C.X. co-first authors.

**Associate Editor:**
Ingo Dreyer

## Abstract

Iron (Fe) is an essential element in plants, involved in numerous metabolic processes including photosynthesis. Its cellular concentration must be regulated accurately to avoid toxicity while meeting metabolic demands. This review explores the distribution, dynamics, and regulation of Fe pools in plant cells, focusing on recent advances in imaging and quantification techniques. We discuss the major Fe compartments—chloroplasts, vacuoles, apoplasts—and their inter-action to maintain Fe homeostasis, as well as novel methodologies like single-cell ICP-MS that have transformed our understanding of Fe localization. By summarizing the current knowledge on intracellular Fe dynamics and the complex interplay between different Fe pools, we provide insights into the mechanisms that underpin Fe regulation in plants, which is crucial for future breeding programs aimed at improving plant resilience and nutritional quality.

## 1. Introduction

As a cofactor of numerous enzymes and a component of redox chains, iron (Fe) is indispensable in all forms of life, playing a role in many essential metabolic processes including photosynthesis in plants. Its concentration in plant cells must be tightly controlled to support metabolic needs without inducing toxicity. Regulating Fe concentration is challenging as plants must adapt to variations of Fe availability in soils. Furthermore, environmental cues such as drought or elevated $CO_2$ also impact Fe content in plants. The adaptive response to Fe-deficiency is decoded and orchestrated by a well characterized cascade of transcription factors that are reviewed elsewhere (Vélez-Bermúdez & Schmidt, 2023). In addition to the regulation of root uptake, identifying and quantifying Fe pools and their dynamics in plant cells is essential to understand the mechanisms of Fe homeostasis and a prerequisite to adapt breeding programs.

As Fe-containing proteins participate in most metabolic reactions, Fe is found in all tissues and cell compartments. Cutting-edge imaging techniques have allowed a lip forward in the description of the main Fe pools in plant cells (Kim et al., 2006; Roschzttardtz et al., 2009). Depending on the tissue, Fe is observed in high amounts in chloroplasts, vacuole, cell wall and nucleolus (Roschzttardtz et al., 2013; Roschzttardtz, Grillet et al., 2011; Roschzttardtz, Séguéla-Arnaud, et al., 2011).

The Fe pools are complex and are comprised of (i) the static pool, which is made of Fe tightly bound to proteins, e.g. in ferritins that sequester Fe in the chloroplasts, or to organic ligands such as Fe-phytate complexes in the vacuole; (ii) The labile pool, chemically uncharacterized yet, which is composed of loosely bound, redox-active and readily available Fe for metabolic reactions (i.e. Fe-citrate, Fe-nicotianamine, Fe-peptides...) (Koppenol & Hider, 2019). Labile Fe must be carefully regulated because of its potential to generate reactive oxygen species. But there is more than meets the eye here since Fe also exists under two redox forms, $Fe^{2+}$ and $Fe^{3+}$, that differ in their biological functions. Thus, techniques devoted to detect and measure Fe in cells must be carefully selected based on the type of iron to be measured.

Traditional tools to study intracellular Fe content rely primarily on ICP-MS-based measurements of bulk samples, which inform on Fe content at the tissue scale. More recently, the advent of dedicated and selective imaging tools such as energy dispersive X-ray analyses (EDX) (Lanquar et al., 2005), synchrotron radiation X-Ray Fluorescence (XRF) (Kim et al., 2006), or the Perls-DAB histochemical staining method (Roschzttardtz et al., 2009) have allowed

precise mapping of Fe at the cell or even subcellular levels, however they fail to give an accurate value of Fe concentration. Recently, single cell approaches have allowed quantifying Fe in isolated cells.

This review attempts to summarize the current knowledge of the major, observable, plant intracellular Fe pools, their distribution and function in the main organelles, with a special emphasis on the methods used to achieve their quantification, and further describes the understanding that emerges on the dynamic interplay between Fe pools of different cellular compartments to attain the set point of Fe concentration in a cell.

## 2. Cell-specific and single cell Fe analyses

Elemental measurement through ICP-MS has been very recently adapted to single cells and revolutionized the quantification of symplastic elements. Two studies report their utilization to determine the ionomic content of cells from different root tissue layers or pollen grains (Giehl et al., 2023; Jiménez-Lamana et al., 2023).

The first study employed cell type-specific ICP-MS, following cell sorting based on fluorescence of tissue-specific reporter genes and established the ionomic profile across Arabidopsis root tissue layers (Giehl et al., 2023). This study revealed the existence of a radial Fe gradient within the primary root, with cellular Fe concentrations increasing from epidermis to endodermis

(Figure 1a). Furthermore, challenging this multi-elemental analysis by exposing plants to various nutritional stresses allowed monitoring nutrient-specific accumulation responses, which follow precise distribution patterns. For example, manganese (Mn) abundance increased in trichoblasts in response to Fe deficiency. The specific Mn retention in this cell-type is a consequence of active loading of Mn into the vacuoles, which protects against Mn overaccumulation in the shoots, emphasizing the role of this cell-type in the complex interplay between Fe and Mn. A specificity of single cell approaches of complex plant tissues is that they require lysis of the cells for sorting, which excludes cell wall-bound fractions. In contrast to radial gradients of symplastic Fe revealed by cell-specific ICP-MS, the majority of labile and total Fe in Arabidopsis roots, shown respectively by Fe imaging using synthetic probes and laser ablation coupled to ICP-MS (LA-ICP-MS), accumulate in the outer cell layers (Alcon et al., 2024; Persson et al., 2016). There, most of labile Fe species accumulate in the apoplastic spaces (Alcon et al., 2024; Roschzttardtz et al., 2013) as discussed below. Together, these independent studies highlight the need for complementary methodologies to evaluate the distribution of the different Fe species and pools.

In contrast to the above-described cell-type ionomic analysis that quantified average Fe concentration from a population of sorted cells, a second study achieved absolute quantification of metal ions in individual and intact cells, by applying single

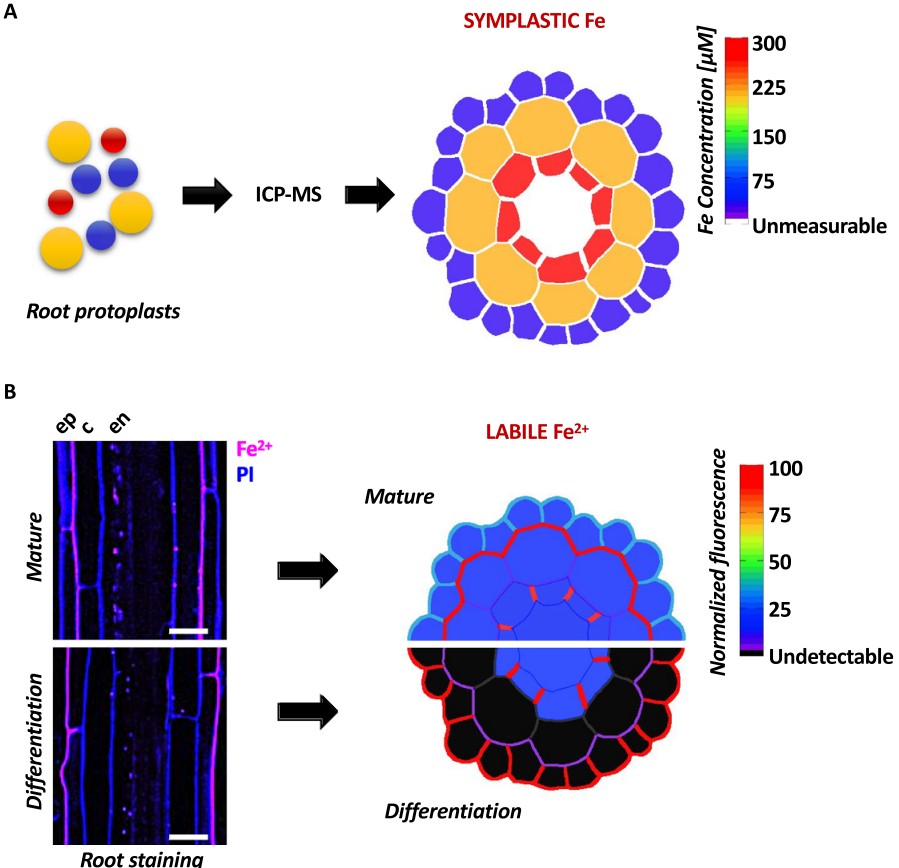

**Figure 1.** Quantitative representation of Fe distribution across different cell layers in the primary root of *Arabidopsis*. (a) Schematic representation of the symplastic Fe concentration measured *via* ICP-MS in tissue-specific cell-sorted root protoplasts, based on data from Giehl et al. (2023). (b) Distribution of labile $Fe^{2+}$ in the Arabidopsis root showing polar localization in the epidermis (based on data from Alcon et al., 2024). Left: Longitudinal confocal laser scanning microscopy images of the differentiation and mature zones of the primary root of Arabidopsis stained with an $Fe^{2+}$-specific fluorescent probe and counter-stained with propidium iodide. The pictures show enrichment of $Fe^{2+}$ in cell walls and opposite polarity in the two root stages. Right: Schematic representation of the distribution and relative quantification of labile $Fe^{2+}$ pools in the same two root regions. Fe levels are represented using a color-coded scale. Magenta LUT: SiRhoNox-1, $Fe^{2+}$; Blue LUT: Propidium iodide, PI. Ep = epidermis; c = cortex; en = endodermis. Scale bar = 20 μm.

cell ICP-MS (SC-ICP-MS) to the pollen grain of *Arabidopsis* (Jiménez-Lamana et al., 2023). Similarly to unicellular organisms, dehiscent pollen grains can be sorted prior to ICP-MS quantification without the need for cell wall lysis, and are therefore easily amenable to elemental quantification at the whole cell level (Jiménez-Lamana et al., 2023; Miyashita et al., 2014; Shen et al., 2019). SC-ICP-MS of pollen grains detected Fe and Mn in the femtogram range ($\approx$6 fg Fe and 7 fg Mn). For comparison, one epidermis protoplast with similar diameter ($\approx$20 μm) contains in average 18 fg of Fe and 5 fg of Mn (Giehl et al., 2023). The study also revealed that a soluble fraction of Fe, but not Mn, is loosely bound to the cell wall and readily released in the medium upon resuspension in water (Jiménez-Lamana et al., 2023). This suggests that in pollen, like in epidermal cells as discussed below, Fe is more likely to accumulate in a labile form at the cell wall, raising the question of the exact nature and physiological role of this Fe pool.

## 3. Chloroplasts as gatekeepers of intracellular Fe balance

Fractionation of leaf cells established that the majority of Fe in plant shoot is localized within chloroplasts, representing approx. 70% of the total leaf tissue content (Shikanai et al., 2003). There, Fe is used in photosynthesis, each electron transport chain containing 22 Fe atoms, as well as for the biosynthesis of heme and Fe–S clusters (Kroh & Pilon, 2020). When Fe supply is abundant or upon senescence, it is sequestered by the storage protein Ferritin within the stroma. Defect of storage increases the pool of free Fe, which has the potential to catalyze reactive oxygen species (ROS) production through the Fenton reaction (Briat et al., 2010). Aberrant ROS are also generated by Fe deficiency as a result of disturbed photosynthesis and impaired antioxidant capacity of the chloroplastic Fe-containing Superoxide Dismutase (FeSOD) (López-Millán et al., 2016). Controlling Fe content in chloroplasts is therefore of utmost importance. The transporters that help Fe cross the chloroplast envelope are not very well defined. Fe presumably enters the chloroplast by the PIC1 permease *via* its interaction with NiCo (Duy et al., 2007, 2011), although several pieces of data point to a more general role in nutrient or even protein import (Eitinger et al., 2005; Teng et al., 2006), and necessitates a prior reduction mediated by the FRO7 ferric reductase at its surface (Jeong et al., 2008). At specific developmental stages such as cotyledons or senescent leaves, Fe is exported from the chloroplast by the YSL4 and YSL6 transporters (Divol et al., 2013). To date, this couple of transporters still lacks an identified substrate, since the presence of the usual substrate of YSL family members, the metal binding-ligand nicotianamine, has not yet been measured in chloroplasts. YSL4 and YSL6 being members of the large family of oligopeptide transporters, their substrate is predicted to be a small peptide with Fe-binding capacities, the nature of which remains to be identified in chloroplasts. With FPN3, which is produced both in mitochondria and chloroplasts, indirect evidences suggest a role in Fe efflux from the two organelles (Kim et al., 2021), but this conclusion would be greatly strengthened by imaging the modification of the organellar Fe pools in the knockout mutant.

In agreement with Fe measurements in isolated chloroplasts, histochemical staining of Fe by the user-friendly Perls-DAB method, which enables to visualize and map static $Fe^{3+}$ in fixed tissues, has established that in *Arabidopsis* leaves, the strongest signal is indeed observed in the chloroplasts, part of it as dot-like structures that were identified as Fe stably bound to Ferritins (Roschzttardtz et al., 2013). Chloroplastic Fe is particularly impacted by insufficient Fe supply. Cellular fractionation and μXRF

of cucumber leaves showed that plants grown under Fe deficiency accumulate undetectable amounts of Fe in chloroplasts (Vigani et al., 2018). Nevertheless, accurate quantification of Fe in a single chloroplast and in response to stresses has not been reported. Furthermore, the full inventory of Fe species in the stroma has not been established. Besides $Fe^{3+}$-Ferritin, which is easy to visualize using histochemical staining of Fe, several labile forms of Fe could co-exist in chloroplasts, including Iron–Sulfur (Fe–S) clusters, the assembly of which takes place for a part in the chloroplast, and labile species with ligands that are knowingly present in chloroplasts such as $Fe^{3+}$-citrate, $Fe^{3+}$-malate and $Fe^{2+}$-ascorbate.

## 4. Iron cycling and storage in seed vacuoles

The transport and storage of Fe in vacuoles of seed tissues is, to date, the best documented example of the dynamic cycling of iron in and out of an organelle. Two decades ago, EDX on *Arabidopsis* seeds provided the first quantitative characterizations of the mineral content of globoïds, which are phytate-rich intra-vacuolar structures known to accumulate high concentrations of phosphate and minerals such as K, Mg, Ca and Fe (Lanquar et al., 2005; Lott et al., 2001). Elemental imaging approaches such as micro Particle-Induced X-ray Emission (μPIXE) and μXRF showed that the globoïds of the aleurone layer of wheat grains are a hot spot for Fe and phosphorus (P), both associated in Fe-phytate complexes, as established by X-ray Absorption Near Edge Structure (XANES) (De Brier et al., 2016; Singh et al., 2014).

Interestingly, Fe isotope uptake coupled with nanoscale-secondary ion mass spectrometry (NanoSIMS) in wheat grains revealed Fe enrichment in endospermic vesicles of different sizes, smaller vesicles devoid of-, and large vesicles bound with phosphorus-oxygen (PO) ligand, likely phytate (Sheraz et al., 2021). The authors were able to capture fusion of new $^{57}$Fe-enriched vesicles with PO-bound vesicles containing only natural (older) $^{56}$Fe isotope. This raises the question whether these Fe-rich vesicles of different composition are eventually shuttled to the vacuoles and/or fuse with the plasma membrane to participate in cell-to cell transport.

Although the speciation of Fe in *Arabidopsis* is similar to wheat, μXRF mapping, as well as histochemical staining of Fe with Perls/DAB, have independently shown that Fe is stored in endodermal cells, a specific cell layer that surrounds the provascular bundles of the embryo (Kim et al., 2006; Roschzttardtz et al., 2009). The VIT1 protein is responsible for the transport of $Fe^{2+}$ in the vacuoles of these cells and this Fe pool builds up all along embryo development as $Fe^{3+}$-phytate (Lanquar et al., 2005; Roschzttardtz et al., 2009). Preventing the upbuilding of this specific vacuolar storage in a *vit1* mutant provokes mislocalization of Fe atoms in vacuoles of the sub-epidermal cells that otherwise store Mn, through the transport activity of the Mn/Fe transporter MTP8 (Eroglu et al., 2017). During germination, Fe ions are retrieved from the vacuoles through the efflux activity of the ferrous iron transporters NRAMP3 and NRAMP4 (Lanquar et al., 2005). Inactivation of *NRAMP3* and *NRAMP4* impaired Fe remobilization and severely compromised growth of the seedlings in Fe-limiting environments (Lanquar et al., 2005; Roschzttardtz et al., 2009). Interestingly, the mis-localization of Fe in the *vit1* mutant also impaired its remobilization, suggesting that, in this case, the capacity to retrieve Fe from sub-epidermal vacuoles is absent (Kim et al., 2006). The fact that iron is transported in and out of the vacuoles as $Fe^{2+}$ but stored as $Fe^{3+}$ implies that a redox

regulation must occur, to ensure proper cycling of Fe atoms in this compartment. The Multidrug And Toxic compound Efflux (MATE) protein AtDTX25 was recently identified as a component of this Fe redox regulation. This membrane protein mediates the transport of ascorbate, a potent $Fe^{3+}$-reducing compound, in vacuoles, and its role is crucial during germination to mediate the reduction of phytate-bound $Fe^{3+}$ and therefore generate $Fe^{2+}$, the substrate of the efflux transporters NRAMP3 and NRAMP4 (Hoang et al., 2021).

## 5. Apoplast as an important buffer compartment of Fe

Apoplastic iron has long been recognized as an important Fe reservoir in plants. In roots of several plant species, apoplastic Fe was shown to constitute approx. 40 to 90% of the total Fe (Bienfait et al., 1985; Giehl et al., 2023; Strasser et al., 1999). Thus, apoplastic Fe is established as one of the major Fe pools in plants. Its size changes dynamically in response not only to Fe availability but also to other nutrients stresses ($NH_4$, P) and to pathogen attack (Liu et al., 2023; Xing et al., 2021). Perls-DAB staining of $Fe^{3+}$ revealed abnormal Fe deposition in the apoplastic compartment of Arabidopsis mutants with impaired Fe transport, such as *irt1 nramp1*, *frd3*, and *ferritin (fer134)* (Castaings et al., 2016; Roschzttardtz et al., 2013; Roschzttardtz, Séguéla-Arnaud, et al., 2011). Though useful for comparative analysis, this method does not provide quantitative measurements of Fe concentrations.

An alternative method consists in the use of fluorescent probes that enable the visualization of metals *in vivo* and offer a more quantitative approach (Alcon et al., 2024; Maniero et al., 2024; Platre et al., 2022). Unlike the Perls-DAB method, recently developed fluorescent Fe probes, applied in combination, allow to simultaneously and distinctively detect the labile Fe redox forms $Fe^{2+}$ and $Fe^{3+}$ (Alcon et al., 2024). Utilization of these probes to investigate the distribution of the labile Fe pools in Arabidopsis root tissues indicated that cell walls, in particular at the periphery of epidermal cells, contains large amounts of labile Fe. Furthermore this study showed that the redox status of Fe is tightly regulated as the root ages since root apex is enriched in $Fe^{3+}$, whereas the elongation and differentiation zones show a complex pattern of both $Fe^{2+}$ and $Fe^{3+}$. In the epidermis, $Fe^{2+}$ ions precisely mapped to the plasma membrane and apoplastic space. In contrast, $Fe^{3+}$ was rather enriched inside the cell, in a large compartment that resembles the central vacuole.

Remarkably, the method from Alcon et al., (2024) uncovered polar distribution of $Fe^{2+}$ ions that are exclusively located at the outer side of the epidermal cells (Figure 1b). Incidentally, the plasma membrane bound-, Fe deficiency-induced, Ferric reductase FRO2 harbours the same polar pattern (Martín-Barranco et al., 2020). This is consistent with the finding of Alcon et al., (2024) that this epidermal $Fe^{2+}$ pool is partially lost in the Fe-deprived *fro2* mutant. Interestingly, the $Fe^{2+}$ polar pattern observed in the differentiation zone switches side in the mature root zone where $Fe^{2+}$ ions accumulate in the inner side of the epidermal cell, i.e. facing the cortex (Figure 1b).

Our knowledge of the role of different cell wall-bound Fe forms is rudimentary. A recent study in Arabidopsis revealed that Fe deficiency triggers galactosylation of the cell wall polymer rhamnogalacturonan-II (RG-II) by inducing the expression of the glycosyltransferase-encoding *Cdi* gene. This, in turn, imposes cell wall modifications that promote desorption of Fe atoms bound to the cell wall and increases tolerance of the plants to Fe deficiency (Peng et al., 2021). Thus upon Fe limiting conditions, Fe can be re-assimilated from apoplastic pools. Nevertheless, labile apoplastic Fe, particularly $Fe^{3+}$, can negatively impact the growth as observed in phosphate (Pi) deficient conditions. In this situation, $Fe^{3+}$ accumulates in the apoplastic spaces of root apical meristem, owing to the activity of the plasma membrane-localized ferroxidase LPR1. There, iron accumulation induces the efflux of malate, through ALMT1, leading to the formation of $Fe^{3+}$-malate complex in the apoplast. This results in the accumulation of ROS that in turn trigger callose deposition and stiffening of the cell walls, slowing down elongation (Müller et al., 2015). To equilibrate the $Fe^{2+}/Fe^{3+}$ ratio in the root tip apoplast, plants have evolved a $Fe^{3+}$ reduction strategy through the production of an ascorbate-dependent ferric reductase related to the CYBDOM family (Clúa et al., 2024; Maniero et al., 2024). Biotic stress conditions, such as pathogen attack can also contribute either to withholding or on the contrary to boosting local apoplastic pools (reviewed in Liu et al., 2023). Thus, apoplastic Fe pools can be either beneficial or detrimental to plant tolerance depending on the form of Fe, type of stress and particular root zone or cell.

## 6. Dynamic interplay between organellar Fe pools

Adequate distribution of Fe between cell compartments ensures plant growth and development under fluctuating Fe conditions. To safeguard such balance when plants face adverse conditions, dynamic fluxes of Fe occur between the different organelles. In Arabidopsis, these fluxes are best illustrated by the communication between vacuolar Fe store and chloroplastic Fe pool since their respective size impacts each other's. When Fe efflux from the chloroplast was shut off as a consequence of inactivating the *YSL4* and *YSL6* genes, vacuolar Fe remobilization by NRAMP3 and NRAMP4 was inhibited (Divol et al., 2013). Conversely, enhancing chloroplastic Fe efflux *via* the overexpression of *YSL4* and *YSL6* led to reduced Fe concentration in the vacuole. Likewise, enlarging the vacuolar Fe pool via either inactivating *NRAMP3* and *NRAMP4* or over-expressing *VIT1*, reduced accumulation of Ferritins, the abundance of which being a readout of the chloroplast Fe status (Ravet et al., 2009).

A similar dialog apparently exists between intracellular and apoplastic Fe pools. Ferritin-less leaves of the Arabidopsis *fer1/3/4* triple mutant accumulate dramatic amounts of Fe in the apoplast of mesophyll cells (Roschzttardtz et al., 2013). It is not clear whether this protective response to the lack of Fe buffering system in the chloroplast results from inhibition of Fe uptake from the intercellular space or from stimulation of a still elusive Fe efflux activity at the plasma membrane. The MCO1 and MCO3 apoplastic ferroxidases, provide another striking example of this dialog. Inactivating *MCO1* and *MCO3* genes, which are highly induced in Fe excess conditions, provoked an overaccumulation of Fe-Ferritin complexes in the chloroplasts, illustrating that the strict control of the Fe redox status is crucial to regulate its uptake from the apoplastic compartment (Brun et al., 2022). Taken together, these findings stress the role of the nature of the intracellular Fe pools and the ongoing communication between them to optimize metabolic functions and plant growth.

## 7. Concluding remarks and open questions

Methodological advances have enabled to map major pools of both tightly bound Fe and labile redox Fe forms within plant cells. Table 1 summarizes the strengths, limitations and the

**Table 1.** Comparison of advanced analytical methods for measuring iron content and dynamics in plant tissues and cells

| | Selectivity | Measurement detection limit | Observed spatial resolution limit (examples) | Sample preparation Destructive/live | Accessibility cost | References |
|---|---|---|---|---|---|---|
| Perls-DAB | Static $Fe^{3+}$ | Qualitative | Tissular to **Subcellular** (nucleus, nucleolus, vacuoles, globoids, Ferritins, cell wall) | Chemical fixation Destructive | **Accessible inexpensive** | Roschzttardtz et al. (2013) |
| SC-ICP-MS | Total iron | **Absolute** < **fg** | **Single cell** (pollen grains, root protoplasts) | Isolated cells Destructive | Accessible expensive | Jiménez-Lamana et al. (2023), Giehl et al. (2023) |
| Fe redox probes SiRhoNox–1 MPNBD | Labile **$Fe^{2+}$** Labile **$Fe^{3+}$** | Semi-quantitative and **dynamic** <1 $\mu$M ($Fe^{2+}$) <70 $\mu$M ($Fe^{3+}$) | Tissular to **Subcellular** (cell wall, vacuole, vesicles) | **No preparation Live tissue** | **Accessible inexpensive** | Alcon et al., 2024 |
| nXRF, $\mu$XRF | Total iron | Semi-quantitative < ppm | Tissular to **Subcellular** (vacuole, nucleus, nucleolus, plastids, cell wall) | **Environmental or cryoprepararation** Destructive | Limited accessibility pending proposal acceptance by Synchrotrons | Kim et al. (2006), Roschzttardtz et al. (2011), Punshon et al. (2012) |
| Nano-SIMS | Total iron | Semi-quantitative <ppm | **Subcellular** (vacuoles, globoids, vesicles) | Lengthy preparation Destructive | Limited accessibility expensive | Sheraz et al. (2021) |
| LA-ICP-MS | Total iron | Semi-quantitative <10 ppm | **Cell-type** (root) | **Freeze-dry** Destructive | Limited accessibility expensive | Persson et al. (2016) |

*Note*: Strengths of each method are indicated in bold.

complementary aspects of these methodologies. However, the identity of the Fe pools in smaller organelles like mitochondria, Golgi apparatus, lyzosomes, or peroxisomes, which all contain important Fe-dependent enzymes, has not yet been deciphered. Visualization and quantification of these pools, an endeavour given how petite these organelles are, could hypothetically be achieved by organellar immunoprecipitation using antibodies targeting protein tags residential to the organelle or organellar subdomains of interest (Drakakaki et al., 2012). With ever-increasing sensitivity of the analytical methods, Fe content in these fractions could be resolved in the future.

Astonishing enrichment of Fe has also been detected in the nucleus and nucleolus of several plant tissues (Hilo et al., 2017; Ibeas et al., 2017; Roschzttardtz, Grillet, et al., 2011), the function of which is obscure. (Montacié et al., 2023) found aberrant ribosomal RNA synthesis in the Fe-deficient nicotianamine triple mutant *nas124*. But it is yet unclear in which form, free or associated to a nucle(ol)ar protein, Fe is present in this compartment.

The wealth of transporters that take in charge the movement of Fe between cell compartments is starting to unveil, yet many of the important ones are still missing from the inventory. Notably, it remains to be established whether Fe efflux activity at the cell surface or vesicular secretion of Fe or both take part in maintaining intracellular Fe concentration. Evidence in favour of the existence of such cellular Fe export mechanisms in plants is still lacking. It is possible that there is sufficient buffering capacity for excess Fe, *via* storage in vacuoles or retention in the apoplasts, to render cellular Fe efflux activity dispensable. The interplay that exists between Fe pools likely involves regulation of intracellular transporter genes, either already known or as yet uncharacterized, as well as a complex set of signals for inter-organellar communication, but this elaborate network has not begun to be explored.

The speciation of Fe is intricately linked with its redox status, since the ligands have very different affinities towards $Fe^{2+}$ and $Fe^{3+}$ (i.e. the stability constant for $Fe^{3+}$-citrate is 8 orders of magnitude higher than that of $Fe^{2+}$-citrate). Therefore, ferric reduction and ferroxidation activities play a central role in the regulation of Fe speciation that will, in turn, affect its mobility and reactivity in the cell. Compared to ferric reduction systems (FRO, ascorbate, ascorbate-dependent CYB-DOM proteins), ferroxidases have received much less attention, despite the fact that they might be as important for Fe distribution in cells. The recent reports on the ferroxidases LPR1 (Naumann et al., 2022; Xu et al., 2022) and MCO1 and MCO3 may likely be the tip of the iceberg in this rather unexplored domain.

There is still a long journey ahead before we can better quantify and localize the numerous Fe pools that are present within a plant cell. However, milestones have been set recently, for example with single cell studies or with selective probes to image the redox forms of Fe. The advent of single cell methods and the newly implemented cell type-specific ICP-MS techniques now offer a robust platform for high-throughput quantitative analysis of cellular metal fingerprints across various tissues, growth conditions and genetic backgrounds. The utilization of fluorescent probes to measure Fe concentrations in plant cells is not flawless. Obstacles that persist include the difficulty to control probe concentration within the cell, or the fact that pH variations that exist between cell compartments or upon stress affect measurement accuracy. Nevertheless, the probes represent an invaluable tool for in the future undertaking the immense task of deciphering Fe content and redox status in all tissues, cells and subcellular compartments. Applications are numerous, including (i) tackling the substrate and activity of candidate transporters, (ii) addressing the dynamics

of Fe by tracking probe fluorescence in living cells. Efforts now need to focus on the development of genetic sensors, which are more efficient than chemical probes to visualize dynamic fluxes of all forms of labile Fe in live tissues and will thus help decipher the hierarchy of Fe shuffling between intracellular pools and the cellular signaling pathways at play.

**Data and coding availability statement.** No data or code were generated as part of this manuscript.

## Acknowledgements

We acknowledge the MRI imaging facility, member of the France-BioImaging national infrastructure supported by the French National Research Agency (ANR-10-INBS-04, "Investments for the future"), and PHIV-La Gaillarde facility.

**Authors contribution.** A.L., T.C.X., S.T. and C.C. wrote the manuscript. T.C.X. made the Figure and Graphical Abstract.

**Financial support statement.** This work was supported by the French Agence Nationale pour la Recherche (Grant Number 2019-CE20-0009), The European Union's Horizon 2020 research and innovation programme (Grant agreement Number 857251-BIOPOLIS), the French Centre national de la Recherche Scientifique (MITI Metallomix Grant 2021 DesciFer), BioCampus Montpellier (Grant 2022 IronSEED).

**Competing interest.** None.

**Open peer review.** To view the open peer review materials for this article, please visit http://doi.org/10.1017/qpb.2025.11.

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
