## [Reviewer Report]

The review has a clear objective: to summarize recent advancements in detecting major iron pools within cells, mapping their distribution, and elucidating their functions in cellular compartments. The manuscript successfully achieves this by presenting cutting-edge methodologies such as single-cell ICP-MS, fluorescent probes, and LA-ICP-MS, which have advanced the detection of symplastic and labile iron, including its oxidation states.

The topic is highly relevant to plant physiology, particularly in understanding iron homeostasis and its role in improving crop resilience and nutritional value. The review is well-organized, with robust conclusions highlighting critical gaps and paving the way for future research. The literature review is comprehensive, referencing significant studies to establish a strong foundation.

Thanks to the implementation of advanced technologies, several significant findings have been highlighted such as: 1) Labile iron is predominantly localized in the apoplast. 2) NanoSIMS has shown iron accumulation in endosperm vacuoles, including vesicle fusion. 3) Substantial amounts of iron have been detected in nuclei and nucleoli.

The manuscript raises important questions for future exploration, such as the role of iron in the apoplast and nucleus, reliable methods for detecting trace levels of iron in organelles, and mechanisms of iron transport across the plasma membrane.

I have only a few comments:

1. The methodology section is thorough and aligns with the study’s objectives, with techniques like single-cell ICP-MS and fluorescence probes highlighted as significant advancements. However, adding a chapter comparing these methods could provide additional value. This section could outline each method’s strengths, limitations, and complementary aspects, enhancing clarity and underscoring how these techniques improve our understanding of intracellular iron dynamics.

2. The manuscript is well-written, but certain sections could be simplified to improve readability using language refinement tools.

3. The sentence regarding iron utilization in the chloroplast (lines 110–111) could perhaps be made a bit more precise. In addition to its role in the electron transport chain and Fe-S protein synthesis, the chloroplast also uses iron for the biosynthesis of other compounds, which might be worth mentioning.

4. Lines 111–112 mention ferritin in the chloroplast stroma but lack information on its developmental stage-specific accumulation. Providing this context would be relevant in understanding iron homeostasis.

5. Although the figure legends provide the methodologies used to detect and quantify iron distribution for both panels A and B, the figure itself only includes the method for panel A, with panel B lacking this information. Including this information on the figure might make it easier to follow.

Overall, this manuscript provides an excellent overview of technological advancements in studying intracellular iron dynamics and makes substantial contributions to the field.

---

## [Reviewer Report]

In this review article, Alexandra Leskova and colleagues present recent advances in our understanding of iron intracellular pools in plants, highlighting major technical advances that enabled those advances. The review also outlines the many remaining challenges and unknows of iron distribution in cells.

The manuscript is authored by a team that is expert of the topic and that contributed many landmark studies in the plant iron homeostasis field. The text is both expert and accessible to the more general reader.

I have only minor comments to further improve the manuscript.

1) In many cases, whole paragraphs or important statements are without citations. I would suggest to include references to support the text. For instance, lines 57-63, line 93, lines 109-115, …

2) Lines 72-74. I would specify which are those two studies (citations + naming more precisely the methods), as it may be unclear down in the text which they are.

3) Lines 162-164. This sentence is unclear to me, as it seems it refers to (an)other species than Arabidopsis, although such species is not specified and the two references at the end of the sentence are about Arabidopsis.

4) The example of chloroplast/vacuole coordinated Fe storage via YSLs and NRAMPs is somehow counterintuitive. Is it correct that it either results in Fe depletion (no YSLs, associated to NRAMPs shutdown) or Fe excess (YSL overexpression associated to reduced vacuolar Fe) in the cytoplasm? One might expect some sort of compensation.

5) Here are a few typos/suggestions :

- line 43 : Fe-proteins  Fe proteins

- line 50 : The  the

---

## [Editor Report]

Dear Cathy et al.,

your manuscript has now been seen by two reviewers. I apologize for the delay, but the last two months of the year are always busy around the world. Both reviewers are very positive and think that the MS makes a substantial contribution to the field. The reviewers mentioned a few points that might help to polish the MS in a minor revision. Please have a look at their comments and see how you can implement their suggestions. Thank you very much for your valuable contribution to the Research Topic “Quantitative approaches to cellular aspects of plant ion homeostasis”.

Best wishes, Merry Xmas and All the Best for the New Year.

Ingo

---

## [Editor Report]

Dear authors,

Thank you for the careful revision of the manuscript. And thanks again for your valuable contribution to the Research Topic “Quantitative approaches to cellular aspects of plant ion homeostasis”. It is highly appreciated.

Best regards, Ingo